# Exploring Metrics to Establish an Optimal Model for Image Aesthetic Assessment and Analysis

**DOI:** 10.3390/jimaging8040085

**Published:** 2022-03-23

**Authors:** Ying Dai

**Affiliations:** Faculty of Software and Information Science, Iwate Prefectural University, 152-52 Sugo, Takizawa 020-0693, Iwate, Japan; dai@iwate-pu.ac.jp

**Keywords:** disentanglement-measure, F-measure, photo score prediction, optimal model, CNN, aesthetics feature

## Abstract

To establish an optimal model for photo aesthetic assessment, in this paper, an internal metric called the disentanglement-measure (D-measure) is introduced, which reflects the disentanglement degree of the final layer FC (full connection) nodes of convolutional neural network (CNN). By combining the F-measure with the D-measure to obtain an FD measure, an algorithm of determining the optimal model from many photo score prediction models generated by CNN-based repetitively self-revised learning (RSRL) is proposed. Furthermore, the aesthetics features of the model regarding the first fixation perspective (FFP) and the assessment interest region (AIR) are defined by means of the feature maps so as to analyze the consistency with human aesthetics. The experimental results show that the proposed method is helpful in improving the efficiency of determining the optimal model. Moreover, extracting the FFP and AIR of the models to the image is useful in understanding the internal properties of these models related to the human aesthetics and validating the external performances of the aesthetic assessment.

## 1. Introduction

With the great growth of digital pictures, many researchers have been interested in exploring the methods of image aesthetic auto-assessment. However, the research on this field is still challenging due to the subjectivity and ambiguity of aesthetic criteria, and the imbalance of the quality distribution. In [1], the authors give an experimental survey about this field’s research. In this paper, besides the discussion of the main contributions of the reviewed approaches, the authors systematically evaluate deep leaning settings that are useful for developing a robust deep model for aesthetic scoring. Moreover, they discuss the possibility of manipulating the aesthetics of images through computational approaches. Recently, facing the issues of the subjectivity and ambiguity of aesthetic criteria, besides predicting the mean opinion score provided by data sets, the approach of predicting the distribution of human opinion scores using a convolutional neural network (CNN) is proposed [2]. Furthermore, the authors use the proposed assessment technique to effectively tune parameters of image denoising and tone enhancement operators to produce perceptually superior results. However, due to the restriction of the CNN, all images are rescaled to square images to feed into the network regardless of their aspect ratios. Following the work in [2], Lijie Wang et al. propose a method of aspect-ratio-preserving multi-patch image aesthetics score prediction [3] in order to reflect the original aspect ratio of information to prediction. In [4], besides keeping the original aspect ratio, the authors propose a spatial attentive image aesthetic assessment model to evaluate the image layout and find spatial importance in aesthetics. In order to improve the learning efficiency during the training process, a multi-patch aggregation method for image aesthetic assessment with preserving the original aspect ratio is proposed [5]. In this method, the goal is achieved by resorting to an attention-based mechanism that adaptively adjusts the weight of each patch of the image. Moreover, in [6], the authors propose a gated peripheral–foveal convolutional neural network. It is a double-subnet neural network. The former aims to encode the holistic information and provide the attended regions. The latter aims to extract fine-grained features on these key regions. Then, a gated information fusion network is employed for the image aesthetic prediction. In [7], the authors propose a novel multimodal recurrent attention CNN, which incorporates the visual information with the text information. This method employs the recurrent attention network to focus on some key regions to extract visual features. However, it has been validated that feeding the weighted key regions to CNN to train the image aesthetic assessment model degrades the performance of prediction according to our preliminary experiments, because the aesthetic assessment is influenced by holistic information in the image. Weakening some regions results in the information degradation for aesthetic assessment.

Furthermore, these methods focus on training the model to predict the distribution of human opinion scores so as to obtain the mean score of the image. So, the personal opinion score that is important in the aesthetic assessment is not reflected. In [8], the authors propose a unified algorithm to solve the three problems of image aesthetic assessment, score regression, binary classification, and personalized aesthetics based on pairwise comparison. The model for personalized regression is trained on the FLICKERAES dataset [9]. However, the ground truth score was set to the mean of five workers’ scores. Accordingly, whether the predicted score embodies the inherently personal aesthetics is not clear. 

On the other hand, some researchers aim at extracting and analyzing the aesthetic features to find the relation with the aesthetic assessment. In [10], the paper presents an in-depth analysis of the deep models and the learned features for image aesthetic assessment in various viewpoints. In particular, the analysis is based on transfer learning among image classification and aesthetics classifications. The authors find that the learned features for aesthetic classification are largely different for those for image classification; i.e., the former accounts for color and overall harmony, while the latter focuses on texture and local information. However, whether this finding is universal needs to be validated further. In [11], besides extracting deep CNN features, five algorithms for handcrafted extracting aesthetic feature maps are proposed, which are used to extract feature maps of the brightness, color harmony, rule of thirds, shallow depth of field, and motion blur of the image. Then, a novel feature fusion layer is designed to fuse aesthetic features and CNN features to improve the aesthetic assessment. However, the experimental result shows that the fusion only improves the accuracy of 1.5% over no-fusion. Accordingly, whether it is necessary to incorporate the inefficiently handcrafted aesthetic features with deep CNN features needs to be investigated.

Recently, the fusion technologies are focused on improving the accuracy of the aesthetics assessment. In [12], the authors introduce a novel, deep learning-based architecture that relies on the decision fusion of multiple image quality scores coming from different types of convolutional neural networks. The experimental results show that the proposed method can effectively estimate perceptual image quality on four large IQA benchmark databases. In [13], the authors propose an aesthetic assessment method, which is based on multi-stream and multi-task convolutional neural networks (CNNs); this method extracts global features and saliency features from an input image. These provide higher-level visual information such as the quality of the photo subject and the subject–background relationship. 

On the other hand, applying image aesthetics assessment to the design field becomes a hot topic in the resent years [14,15,16,17]. For example, in [14], the impact of cover image aesthetics on content reading willingness is analyzed. In [15], the food images are assessed by learning its visual aesthetics. In [16], the abstract images are generated by using correction structure with the aesthetics. 

However, the above papers ignore the fact that the distribution of samples against aesthetic scores in a data set is highly non-uniform [18,19]. Most images in the dataset assessed by a professional photographer have the score of 4, and about 85% of the images in the dataset concentrate on the scores of 3 to 5. The score classification model could be overwhelmed by those samples in the majority classes if the parameters are learned by treating all samples equally. Accordingly, in order to solve the problem of the imbalanced classification in image aesthetic assessment, the papers of [18,19] propose a mechanism of repetitively self-revised learning (RSRL) to train the CNN-based image score prediction model on the imbalanced score data set. As RSRL, the neural networks are trained repetitively by dropping out the low likelihood photo samples at the middle levels of aesthetics from the training data set based on the previously trained network, and the optimal model that has the highest value of F-measure is selected from them. The experimental result presents that this model outperforms the model trained without RSRL. However, because the F-measure is dependent on the test dataset, whether it reflect the internal property of the model is not clear. 

In this paper, we focus on the issue of CNN-based RSRL to explore suitable metrics, including one reflecting the internal properties of the model, to establish an optimal model for image aesthetic assessment. Moreover, the learned feature maps of the model are utilized to define the first fixation perspective (FFT) and the assessment interest region (AIR), so as to analyze whether the aesthetics features are learned by the optimal model. 

In details, the main contributions of this paper are summarized as follows.

For RSRL, besides the external F-measure, an internal metric called the disentanglement-measure (D-measure), which measures the degree of disentanglement of final FC layer nodes, is defined.By combining the F-measure with the D-measure to obtain an FD measure, an algorithm of determining the optimal model from many re-trained models generated by RSRL is proposed, while these models are score prediction models.The effectiveness of the proposed method is validated by comparing the performances of many re-trained models with different CNN structures.The FFP and the AIR of the model to the image are defined by the activated feature maps. It is found that analyzing the FFP and AIR of the images is useful in understanding the internal properties of the model related to the human aesthetics and validating its external performances of the aesthetic assessment.

The reminder of this paper is organized as follows. Section 2 introduces the D-measure and Section 3 describes how to establish the optimal model for image aesthetic assessment via RSRL. Section 4 explains the method of extracting FFP and AIR. Section 5 gives the experimental results and analyzes the effectiveness of the proposed methods.

## 2. Disentanglement Measure

The approach of RSRL is proposed in [18,19]. In this paper, in order to solve the data imbalance in training the model for photo aesthetic assessment, the author focuses on conducting repetitive self-revised learning (RSRL) to retrain the CNN-based photo score prediction model repetitively by transfer learning, so as to improve the performance of imbalanced classification caused by the highly non-uniform distribution of training samples against scores. For RSRL, the network is trained repetitively by dropping out the low likelihood photo samples with mid-level scores from the training data set based on the previously trained model. Then, as the photo score predictor, the optimal model is determined from the many re-trained models according to the F-measure. However, the F-measure is an external measure of the model. Whether it reflects the intrinsic property of the model is unclear. Accordingly, this paper conducts a new internal measure called the disentanglement-measure (D-measure), which measures the degree of disentanglement of final FC layer nodes of CNN. The idea behind it is that the nodes of the final FC layer should be disentangled for the accurate classification. However, it is obvious that the adjacent score classes are relevant. So, it is impossible for score classification to disentangle the final FC layer nodes of CNN completely. Then, the D-measure of these nodes against their weights with the nodes of the last FC layer is defined. In the following, we describe how to define the D-measure in detail. Figure 1 shows the situation of the last two FC layers of CNN.

The flowchart of defining the D-measure is shown in Figure 2.

Let the node of the final FC layer be $fc_{j}^{N}$, the node of the penultimate FC layer be $fc_{i}^{N-1}$, and the weight between $fc_{i}^{N-1}\;\mathrm{and}\;fc_{j}^{N}$ be $w_{ij}^{N-1,N}$. Furthermore, the number of $\;fc_{j}^{N}$ is J, and the number of $\;fc_{i}^{N-1}$ is *I*. The details of calculating the D-measure are explained as the following steps.


Normalizing $w_{ij}^{N-1,N}$ as $w1_{ij}^{}$;Calculating the correlation matrix of $fc_{j}^{N}$ based on $w1_{ij}^{}$;


(1)$$R=\frac{1}{I-1}w1_{ij}^{\prime }\times w1_{ij\;\;}^{},\;i\in \left[1,I\right],\;\mathrm{and}\;j\in \left[1,J\right]$$
where R is a matrix of J×J.

Calculating and sorting the eigenvalues in a descending order, denoted $eign_{m}$, and obtaining the corresponding eigenvectors, denoted $eign\_v_{m}$;Calculating the factor loading of factor *m* (latent variable) against $fc_{j}^{N}$, *j* ∈ [1, J];

(2)$$fl_{m,j}=\sqrt{eign_{m}}\;\times eign\_v_{m,j},\;m\in \left[1,M\right]$$
where M indicates the number of factors.

Calculating the two-norm of factor loadings against the two nodes *j*1 and *j*2 of $fc_{j}^{N}$;



(3)
$$dis_{j1,j2}=2-norm\left(fl_{m,j1}-fl_{m,j2}\right)$$



The larger the value is, the more the two nodes *j*1 and *j*2 are leaved.

Calculating the minimum of two-norm regarding one node *j* to all of the other nodes *jj* of $fc_{j}^{N}$;



(4)
$$dis\_min_{j}=min\left(dis_{j,jj}\right)$$



Calculating the mean of the minimum of two-norm regarding all of the nodes of $fc_{j}^{N}$;

We can see that the mean of $dis\_min_{j}(j\in J$) reflects the dispersion of all nodes. The larger value indicates that the nodes are more scattered. 

Defining the D-measure

We define the D-measure of the nodes of $fc_{j}^{N}$ as the following expression.
(5)$$D-\mathrm{measure}=\mathrm{mean}\left(dis\_min_{j}\right)$$

In the next section, the role of D-measure in establishing an optimal model for image aesthetic assessment via CNN-based RSRL will be discussed.

## 3. Establishing an Optimal Model for Image Aesthetic Assessment via RSRL

An improved mechanism of CNN-based RSRL is shown in Figure 3. 

The approach of CNN-based RSRL is to drop out the low likelihood samples of the majority classes of scores repetitively, so as to ameliorate the invasion of these samples to the minority classes and prevent the loss of the samples with discriminative features in the majority classes. In this process, the previous model is re-trained by transfer learning again and again. Accordingly, many re-trained models are generated with RSRL. Then, the optimal model is determined among these models based on the F-measures [10]. However, the F-measure is dependent on the test dataset reflecting the external property of the models, so whether it embodies the internal property of the model is not clear. In this purpose, we introduce the D-measure with F-measure to determine the optimal model because the D-measure reflects the internal property of the classification. In detail, let the F-measure of class *j* be $F_{j}$ and the total F-measure of the classes be F_all. F_all is calculated by Equation (6).
(6)$$F\_\mathrm{all}=\overset{J}{\underset{j=1}{\sum }}F_{j}$$

Then, the measure that aggregates F_all and D-measure is defined by Equation (7). Here, the values of F_all and D-measure are normalized.
(7)$$\mathrm{FD}=w_{1}F\_\mathrm{all}+w_{2}D-\mathrm{measure}$$
where $w_{1}$ and $w_{2}$ are set as 0.5, respectively.

It is assumed that the optimal model should be one that has the maximal value of F_all among the re-trained models, and the value of its FD is larger than a threshold *T*. Then, the algorithm selecting the optimal model is expressed in Expression (8).
(8)$$L_{optimal}=\{\begin{array}{c} \underset{l\in L}{argmax}{F\_\mathrm{all}}_{l}\\ FD_{L_{optimal}}>T \end{array}$$
where the $L_{optimal}$ indicates the index of the optimal model, and *l* and *L* indicate the index and the number of the re-trained models, respectively. In this paper, *T* is set to 0.95, which will be explained in Section 5.1.

On the other hand, if only the metrics of F_all or D-measure are used, the corresponding optimal model is obtained by Expressions (9) or (10), respectively.
(9)$$L_{optimal}^{F\_\mathrm{all}}=\underset{l\in L}{argmax}{F\_\mathrm{all}}_{l}$$
(10)$$L_{optimal}^{D}=\underset{l\in L}{argmax}D_{l}$$

Then, scores of images can be predicted by the combination of models $L_{optimal}^{F\_\mathrm{all}}$ and $L_{optimal}^{D}$. The formulation for predicting the score is expressed by Equation (11)
(11)$$\mathrm{score}=\underset{j\in J}{argmax}(w_{1}fc_{j}^{F\_\mathrm{all}}+w_{2}fc_{j}^{D})$$
where $fc_{j}^{F\_\mathrm{all}}$ and $fc_{j}^{D}$ indicate the sigmoid values of the final FC layers regarding the models $L_{optimal}^{F\_\mathrm{all}}$ and $L_{optimal}^{D}$, respectively. $w_{1}$ and $w_{2}$ are set as 0.5, respectively.

The effectiveness of utilizing the D-measure to obtain $L_{optimal}^{D}$ in the prediction is validated by experiments. The results and analysis will be shown in Section 5.

## 4. Extracting FFP and AIR

Although there are composition attributes for taking good photos, such as rule of third and depth of field, people are more likely to concern the first fixation perspective (FFP) and the relation with other elements when enjoying photos, which are considered to be the assessment interest region (AIR). Accordingly, for a CNN-based photo aesthetic assessment model, it is supposed that the most activated feature map should be related to the FFP of the image, and the sum of feature maps should be related to the AIR. So, the model’s FFP and AIR of the image could be acquired by the following calculation.

Obtaining the most activated feature map and the sum of feature maps of the final convolutional layer of CNN regarding an image I(x,y) and normalizing and resizing these to the size equal to the I, where x and y indicate the coordinates of the pixel, respectively;

Let a feature map of the final convolutional layer be $FM^{p}\left(x,y\right)$, its index be *p*, the number of feature maps be *P*, and the index of the most activated feature map be $p_{max}$.
(12)$$p_{max}=\underset{p\in P}{argmax}FM^{p}\left(x,y\right)$$

Accordingly, the most activated feature map should be $FM^{p_{max}}\left(x,y\right)$. Then, let the sum of feature maps be $FM_{sum}\left(x,y\right)$.
(13)$$FM_{sum}\left(x,y\right)=\overset{P}{\underset{p=1}{\sum }}FM^{p}\left(x,y\right)$$

Extracting FFP and AIR;

Let FFP be represented by FFP(x,y), and AIR be represented by AIR(x,y), then,
(14)$$\mathrm{FFP}\left(x,y\right)=I\left(x,y\right)\times FM^{P_{max}}\left(x,y\right)$$
(15)$$\mathrm{AIR}\left(x,y\right)=I\left(x,y\right)\times FM_{sum}\left(x,y\right)$$

Figure 4 shows the examples of FFP(x, y) and AIR(x, y) of an image, which are calculated based on the learned feature maps of the model with three 1 × 1 convolution layers. 

It seems that the highlighted regions of (b) and (c) are close to the human’s perception regarding FFP and AIR when enjoying an image. 

In Section 5, we will show whether the optimal model determined by (8), (9), or (10) could learn the FFP(x, y) and the AIR(x, y), which are close to the human’s aesthetic perceptions.

## 5. Experimental Results and Analysis

### 5.1. Establishing Optimal Model

In this paper, we construct three kinds of CNN models with different structures to validate whether the D-measure metrics are helpful for determining the optimal model. These three kinds of CNN structures are fine-tuned AlexNet, changed AlexNet, and new designed only 1 × 1 convolutions CNN, which are presented below. Although there are many popular pre-trained models such as ResNet, we think that it is suitable to select AlexNet as a representative to validate the effectiveness of the proposed method [20].
Type a: Fine-Tuned AlexNet1--end-3 layers: transferring 1--end-3 of AlexNetend-2 layer ‘fc’: 8 fully Connected layer, each corresponding to a score class of 2–9end-1 layer ‘softmax’: Softmaxend layer ‘classoutput’: Classification Output
Type b: Changed AlexNet1--end-9 layers: transferring 1--end-5 of AlexNetend-8 layer ‘batchnorm_1’: Batch normalization with 4096 channelsend-7 layer ‘relu_1’: ReLUend-6 layer ‘dropout’: 50% dropoutend-5 layer ‘fc_1’: 32 fully connected layerend-4 layer ‘batchnorm_2’: Batch normalization with 32 channelsend-3 layer ‘relu_2’: ReLUend-2 layer ‘fc_2’: 8 fully Connected layer, each corresponding to a score class of 2–9end-1 layer ‘softmax’: Softmaxend layer ‘classoutput’: Classification Output
Type c: Only 1 × 1 convolutions CNN1 layer ‘imageinput’: 227 × 227 × 3 images with ‘zerocenter’ normalization2 layer ‘conv_1’: 94 1 × 1 × 3 convolutions with stride [8 8] and padding [0 0 0 0]3 layer ‘batchnorm_1’: Batch normalization with 94 channels4 layer ‘relu_1’: ReLU5 layer ‘conv_2’: 36 1 × 1 × 94 convolutions with stride [4 4] and padding [0 0 0 0]6 layer ‘batchnorm_2’: Batch normalization with 36 channels7 layer ‘relu_2’: ReLU8 layer ‘conv_3’: 36 1 × 1 × 36 convolutions with stride [1 1] and padding [0 0 0 0]9 layer ‘batchnorm_3’: Batch normalization with 36 channels10 layer ‘relu_3’: ReLU11 layer ‘fc_1’: 36 fully connected layer12 layer ‘fc_2’: 8 fully Connected layer, each corresponding to a score class of 2~913 layer ‘softmax’: Softmax14 layer ‘classoutput’: Classification Output

By using transfer learning, each of these CNNs are re-trained 29 times iteratively via RSRL on the xiheAA dataset [10], which was mentioned in the section introduction. Four out of the five samples are randomly selected as the training dataset, and the remaining sample serves as the validation dataset. Here, it is indicated that the 29 times is only an example for showing how to select the optimal model from the retrained models.

Figure 5, Figure 6 and Figure 7 show the values of D-measure, F_all, and FD of 29 re-trained models regarding three kinds of CNNs on the validation dataset, respectively. The values of D-measure and F_all are all normalized.

From the results shown in these figures, the optimal models of type a, type b, and type c should be model 20, model 28, and model 2 by means of Expression (8). Moreover, we can see that for the type b and type c; the optimal models have the maximal FD with the maximal F_all. Simultaneously, these models have the comparatively high values of D-measure. The FD measures can reach the values of more than 0.98.

For the type a, although the re-trained model with maximal FD is the model with the maximal D-measure, the model 20 having the second largest FD is the one with the maximal F_all. Moreover, the FD of that is 0.96, which is larger than the threshold 0.95. So, this model should be selected as the optimal model based on Expression (8).

Now, let us analyze the effectiveness of Expression (8) in determining the optimal model from the many re-trained models. For the various re-trained models, we notice that the models with the maximal D-measure values are not necessarily ones with maximal F_all values, and vice versa. However, the model with maximal F_all should have the high D-measure value. This means that although the D-measure reflects the dispersion of the nodes, it cannot be said that it embodies the external property of the classification regarding the models. However, the model having the maximal F_all possesses the comparatively high D-measure. So, we can say that the D-measure measuring the internal classification property of the model is a necessary condition for the good classification, but it is not a sufficient condition. Accordingly, combining the metrics of F_all with the D-measure to obtain the metrics of FD is helpful for determining the optimal model. The corresponding algorithm is expressed by (8). If the value of FD reaches a threshold, for example, 0.95, and the F_all is maximal at this iteration, the RSRL can be stopped, and the current re-trained model is used as the optimal model. The benefits of doing so can improve the efficiency of determining the optimal model compared with the only F-measure-based RSRL [18,19]. For example, in the case of Figure 6, RSRL can be stopped at the second iteration, while the FD is larger than 0.95. 

Next, the CUHK-PQ dataset [1] is used for the out-of-distribution validation. The CUHK-PQ dataset contains 10,524 high-quality images and 19,166 low-quality images. So, the images predicted of having the score of less than 5 are assigned to the low class, while the others are assigned to the high class. Figure 8 shows the experimental results.

The values of vertical axis indicate the F-measure’s average values regarding the different optimal models of types a, b, and c. The yellow bar (FD) represents the optimal model determined by Expression (8). The red bar (Fall) and blue bar (D) represent the optimal models determined by Expressions (9) or (10), respectively. It is obvious that the heights of the red bars and yellow bars are the same, but the height of the blue bar is a little low, although it is very close to the yellow bar. The purple bar (Fall + D) represents the optimal model determined by Expression (11). It is observed that the height of the purple bar regarding type a is slightly lower than the yellow bar; that regarding type b is slightly higher than the yellow bars; but, that regarding type c is obviously higher than the yellow bar.

The above analyses further show that the D-measure is a necessary condition for the good classification, but it is not a sufficient condition. However, although the performance of the optimal model determined by the FD measure, which is the aggregation of F_all and the D-measure, is almost as same as the one determined by the metrics of F_all, the efficiency of determining the optimal model from many re-trained models can be improved.

On the other hand, it is observed that the models of type b have about a 10% higher average F-measures than the other two types of models. That is, inserting new layers into the FC section of CNN can improve the performance of predicting scores. However, it is more interesting that the average F-measure of the optimal model of type c determined by FD is almost as same as that of type a. Moreover, the average F-measure of type c obtained by the aggregation of models $L_{optimal}^{F\_all}$ and $L_{optimal}^{D}$ is 1.8% higher than that of type a. Accordingly, we could deduce that the newly designed CNN with only three layers 1 × 1 convolutions via RSRL on a small dataset could outperform the fine-tuned pre-trained CNN, such as AlexNet. Moreover, the size of type c is only about 1/620 of type a and type b. So, in the view of the ratio of cost-effectiveness, the only 1 × 1 convolution CNN may be better as a score prediction model, although the type b, which is the changed AlexNet, is better in the view of F-measure.

### 5.2. Extracting FFP and AIR

Figure 9 is an example of the original image. Figure 10 and Figure 11 show the FFP(x, y) and AIR(x, y) of this image, which are acquired based on (14) and (15) using the FD-determined and the D-determined optimal models of type a, b, and c, respectively. The FD-determined optimal model means that the optimal model is determined by Expression (8), and the D-determined optimal model means that the optimal model is determined by Expression (10). This image is from the CUHK-PQ dataset with high-quality labeling. The predictions of the three FD-determined models are score 5, score 7, and score 5, respectively; the predictions of the three D-determined models are score 7, score 7, and score 5, respectively.

From the results, we can see that for all of the types, two kinds of AIR(x, y) are almost the same. In the case of types b and c, the highlighted region that is considered as the AIR is the one having the hand with goldfish, while it is very close to the human aesthetics assessment when enjoying the photo. However, in the case of type a, the highlighted region as AIR is the surrounding area that seems not to be consistent with the human aesthetic habit, although the predicted result is right. These observations may indicate that fine-tuning a pre-trained CNN on the small score dataset cannot make the re-trained model learn the aesthetic features, but it may be possible to make the CNN models learn the deep aesthetic features by changing the pre-trained CNN structure or training a new multi-layer only 1 × 1 convolutions CNN. 

However, the two kinds of FFP(x, y) are not the same for the models of type b, although they are the same for the models of type a and type c. We think that the FFP of the image should be the area of the goldfish for the human perception. The two optimal models of type a and type c indeed learn the elements of FFP as the human; however, for type b, the FD-determined optimal model extracts the surrounding of the hand as FFP, although the D-determined optimal model extracts the goldfish area. 

On the other hand, it is observed that the highlighted regions of FFP(x, y) and AIR(x, y) obtained by the FD-determined optimal model is stronger, and the highlighted area is larger. It may be the reason that the FD-determined optimal model either outperforms or rivals the D-determined optimal model, because the aesthetics assessment to an image is related to the element composition in the image but not the isolated object.

Figure 12 is another example of the original image. Figure 13 and Figure 14 show the FFP(x, y) and AIR(x, y) of this image, using the FD-determined and the D-determined optimal models of type a, b, and c, respectively. This image is from the CUHK-PQ dataset with a low-quality labeling. The predictions of the three type’s models are score 4, score 2, and score 3, respectively.

Similar to the above analysis, we can see that for all of the types, two kinds of AIR(x, y) are almost the same. In the case of types b and c, the highlighted region that is considered as the AIR has the weeds that result in the low assessment, while it is very close to the human’s assessment to this photo. However, in the case of type a, the highlighted region is unclear. This may be the reason that the prediction is score 4. 

On the other hand, for all of the types, the two kinds of FFP(x, y) are almost the same, too, although these are different for different types. We think that the FFP of the image should be the area of the flower for the human perception. The two optimal models of type c indeed learn the elements of FFP as the human; however, for type a and type b, two optimal models extract the surrounding of the flower as FFP. 

Figure 15 is the third example of the original image. Figure 16 and Figure 17 show the FFP(x, y) and AIR(x, y) of this image, using the FD-determined and the D-determined optimal models of types a, b, and c, respectively. This image is from the CUHK-PQ dataset with a high-quality labeling. The predictions of three FD-determined models are score 4, score 4, and score 3, respectively; the predictions of three D-determined models are score 4, score 4, and score 4, respectively. This means that the prediction is not consistent with the ground truth. The image with a high-quality label is assigned to the low-quality class, especially in the case of type c. 

For all of the three types, we can see that the FFP(x, y) and AIR(x, y) do not meet the rule of thirds in photography. This may be the reason that it results in the incorrect prediction. 

In fact, about 300 images selected randomly from the CUHK-PQ dataset and the website are used to investigate the properties of their FFP(x, y) and AIR(x, y). Although they are various for different images, for almost of the images, it is observed that the optimal models of type b and type c, especially type c, can learn FFP and AIR, which are closer to the human aesthetics. It seems that extracting the FFP and AIR of the image can help in understanding the internal properties of the model related to the human aesthetics and validating its external performances of the aesthetic assessment, although it is necessary to do more experiments to validate these observations.

## 6. Conclusions

To establish an optimal model for photo aesthetic assessment, in this paper, an internal metric called the D-measure, which reflects the disentanglement degree of the final layer FC nodes of CNN was introduced. By combining the F-measure with the D-measure to obtain an FD measure, an algorithm of determining the optimal model from many photo score prediction models generated by a CNN-based RSRL was proposed. Furthermore, the FFP and the AIR of the models to the image were defined and calculated. Compared with the only F-measure-based RSRL, using the FD measure to determine the optimal model from the re-trained models is helpful in improving the efficiency of selecting the optimal model, although the D-measure is only the necessary condition for a model having the high F-measure. 

Furthermore, extracting the FFP and AIR of the models to the image can help in understanding the internal properties of these models related to the human aesthetics and validating its external performances of the aesthetic assessment. The experimental results show that the optimal models of type b and type c, especially type c, can learn FFP and AIR, which are closer to the human aesthetics.

In the next work, it is necessary to do more experiments to validate the above observations and the effectiveness of the proposed method.

On the other hand, we think that it is possible to apply the proposed method to the different domains, which have the open issues of the imbalance classification.

## Figures and Tables

**Figure 1 jimaging-08-00085-f001:**
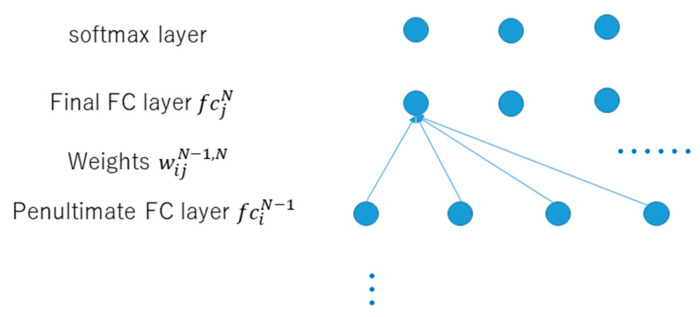
Last two Full connection (FC) layers of convolutional neural network (CNN).

**Figure 2 jimaging-08-00085-f002:**
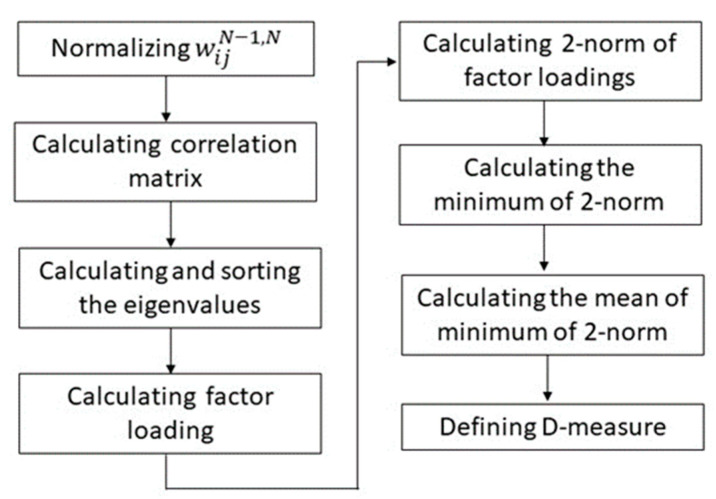
Flowchart of defining D-measure.

**Figure 3 jimaging-08-00085-f003:**
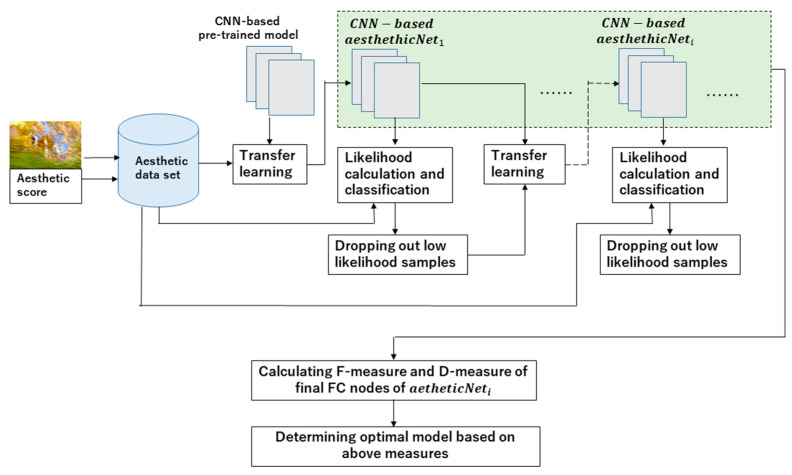
Improved CNN-based RSRL.

**Figure 4 jimaging-08-00085-f004:**
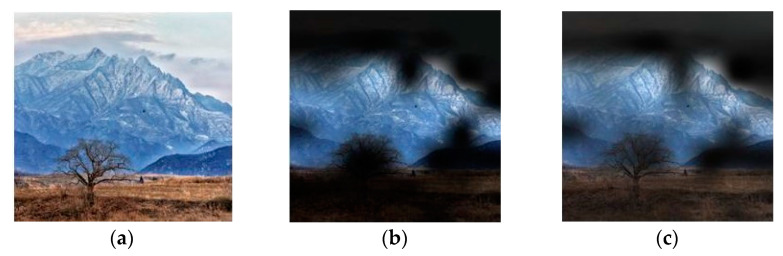
Examples of FFP(x, y) and AIR(x, y). (**a**) Original image; (**b**) FFP(x, y); (**c**) AIR(x, y).

**Figure 5 jimaging-08-00085-f005:**
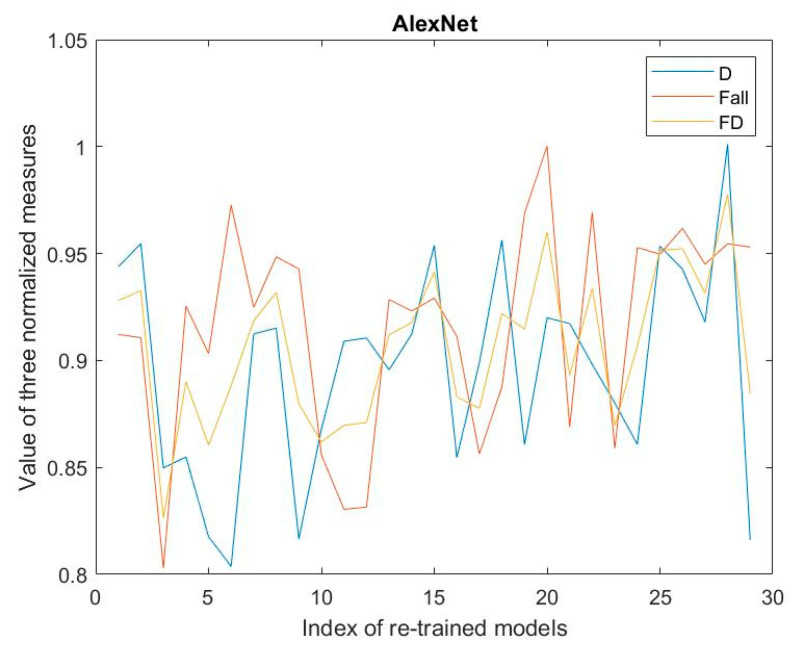
Results regarding fine-tuned AlexNet.

**Figure 6 jimaging-08-00085-f006:**
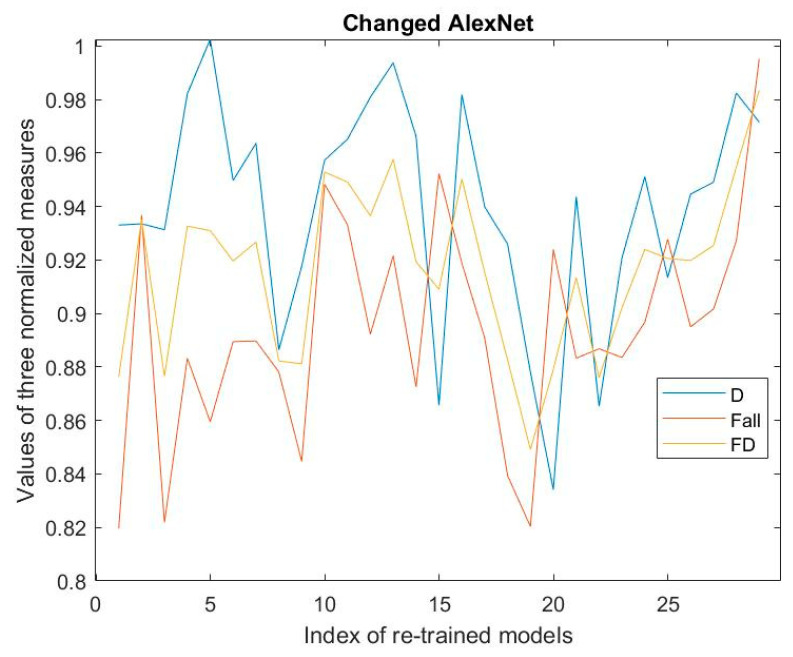
Results regarding changed AlexNet.

**Figure 7 jimaging-08-00085-f007:**
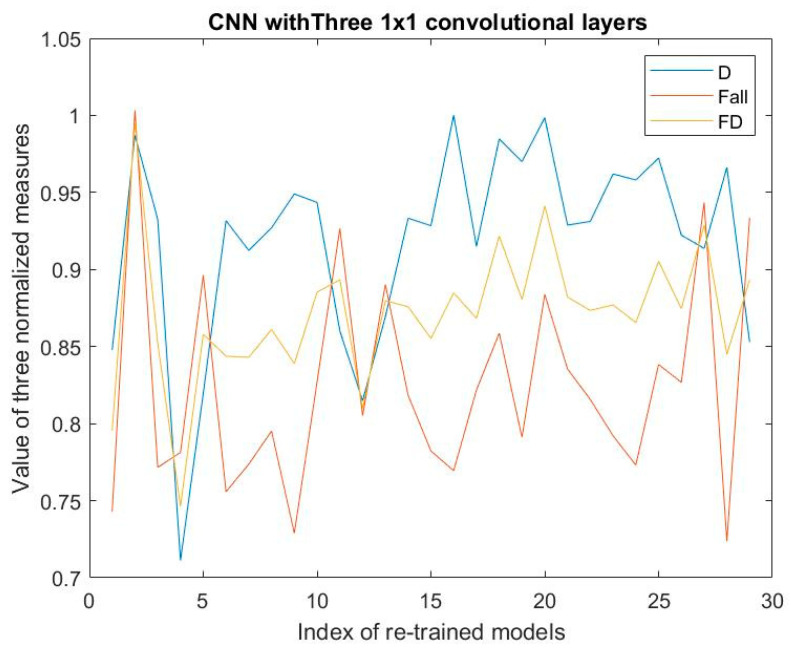
Results regarding only 1 × 1 convolutions’ CNN.

**Figure 8 jimaging-08-00085-f008:**
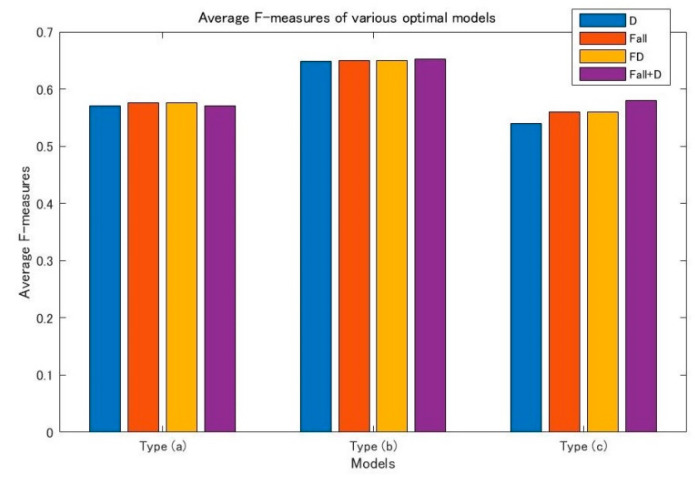
Average F-measures of different optimal models.

**Figure 9 jimaging-08-00085-f009:**
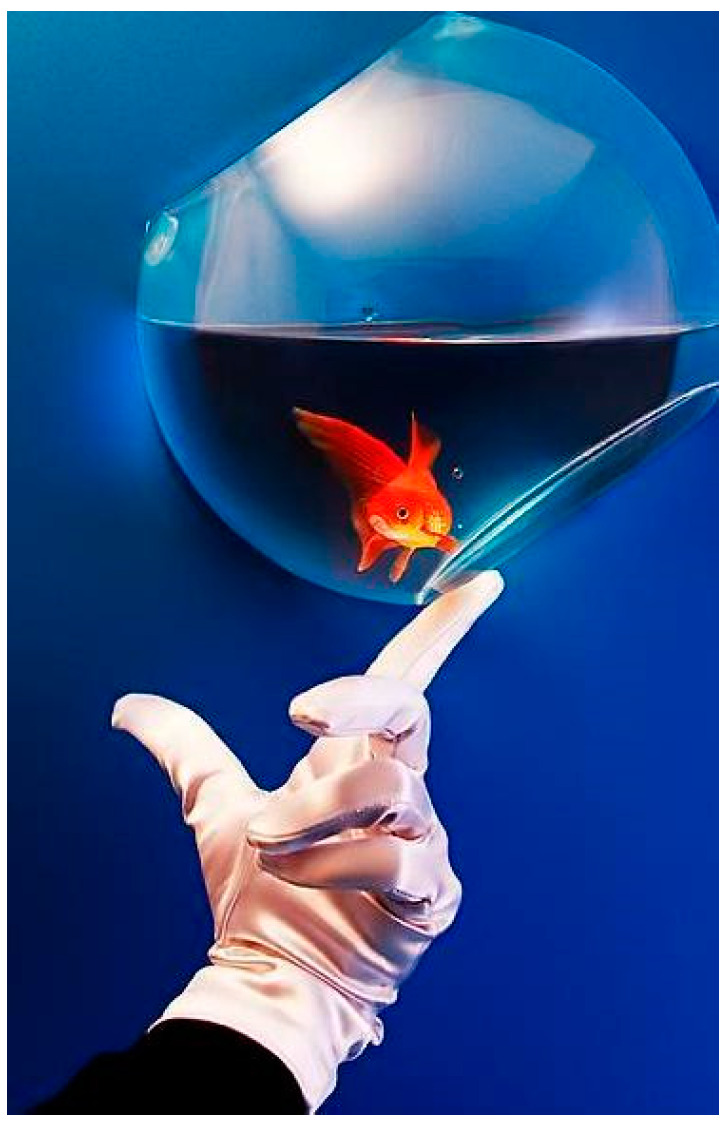
Original image.

**Figure 10 jimaging-08-00085-f010:**
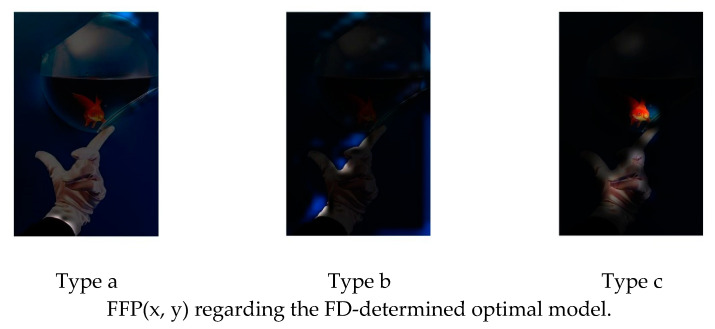
FFP(x, y) of Figure 9.

**Figure 11 jimaging-08-00085-f011:**
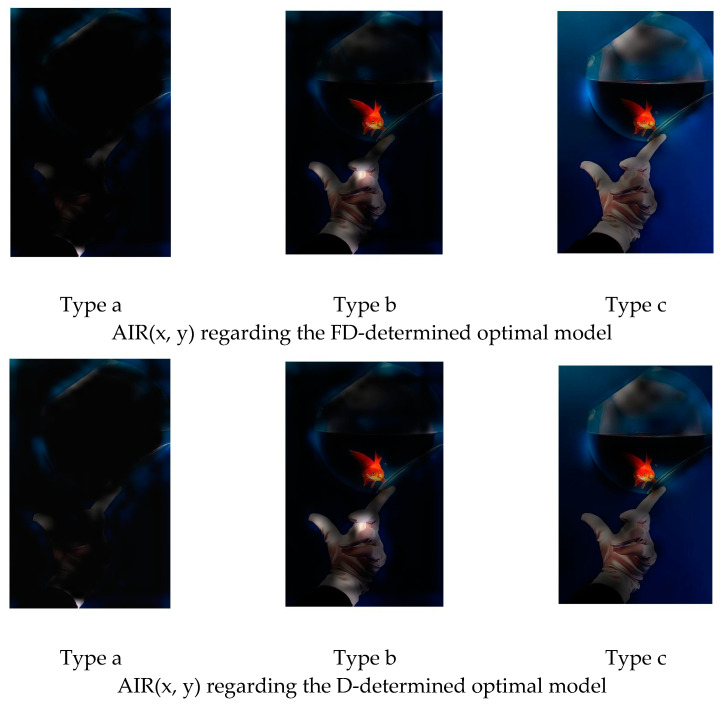
AIR(x, y) of Figure 9.

**Figure 12 jimaging-08-00085-f012:**
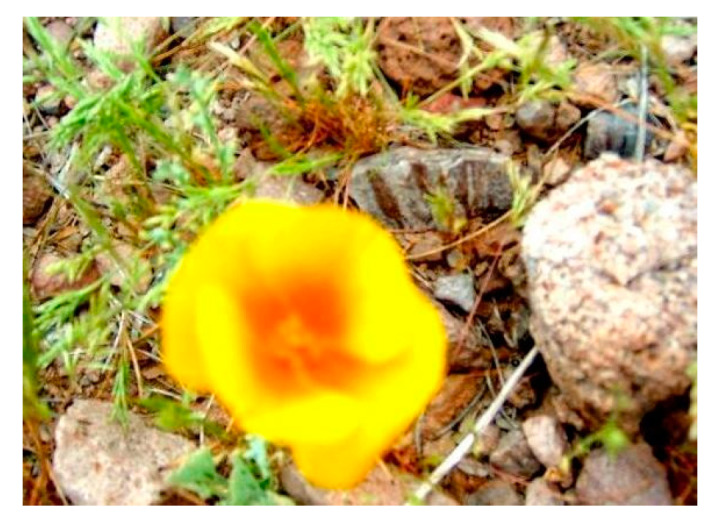
Original image.

**Figure 13 jimaging-08-00085-f013:**
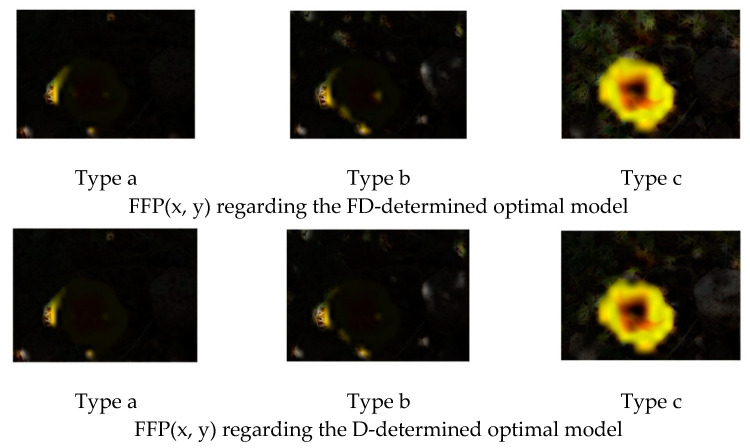
FFP(x, y) of Figure 12.

**Figure 14 jimaging-08-00085-f014:**
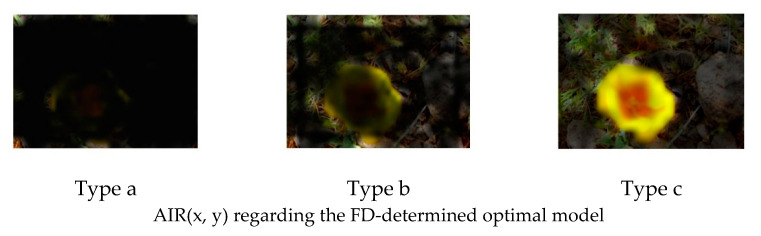
AIR(x, y) of Figure 12.

**Figure 15 jimaging-08-00085-f015:**
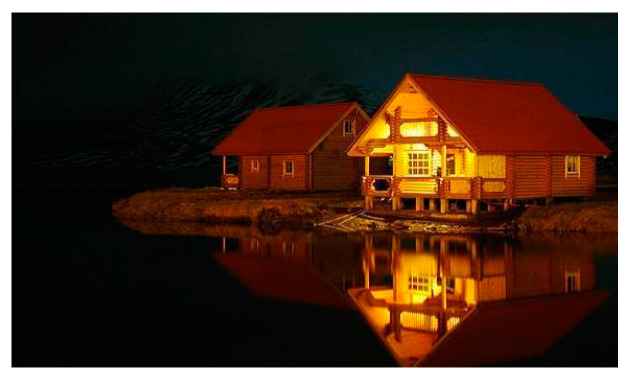
Original image.

**Figure 16 jimaging-08-00085-f016:**
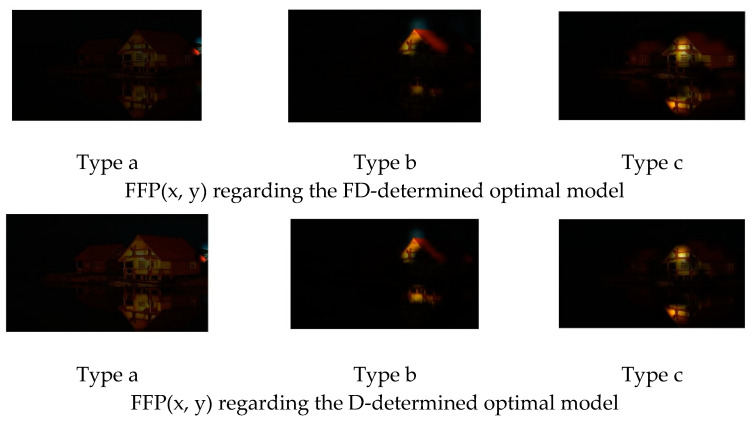
FFP(x, y) of Figure 15.

**Figure 17 jimaging-08-00085-f017:**
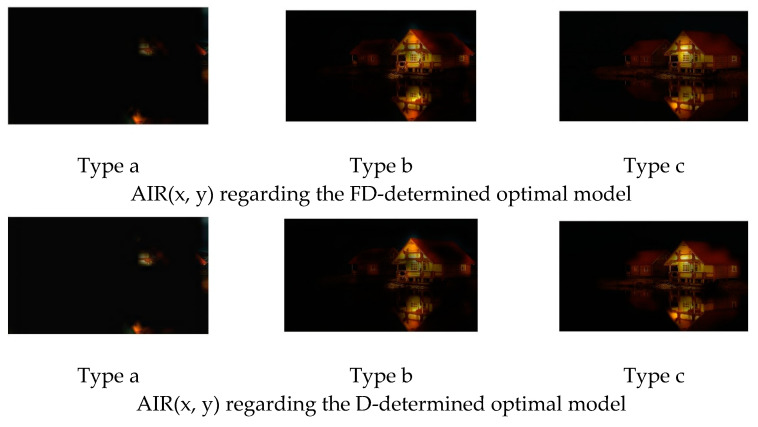
AIR(x, y) of Figure 15.

## Data Availability

Not applicable.

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
