# Peer review of "Exploring Metrics to Establish an Optimal Model for Image Aesthetic Assessment and Analysis"

_2313-433X, 2022, doi:10.3390/jimaging8040085_

Round 1

Reviewer 1 Report

The paper is presenting an important topic, I have  some comments to improve it.

1- author has to mention there is a preprint for the paper.

2- More experiments are required to validate the proposed method such as applying it on different models such as ResNet with standard datasets.

3- comparison with previous methods 

4- elaborate more on the novelty of the proposed method 

5- elaborate more on the benefit of the proposed method such as in terms of performance 

6- code , demo are important to be public 

7- More recent  references are required,  I would suggest the following,  update other references 

" Review of deep learning: concepts, CNN architecture, challenges, applications, future directions "

Author Response

  • author has to mention there is a preprint for the paper.

Answer:

I have uploaded the paper on the platform with the link as the following.

https://www.preprints.org/user/home/submissions/pending

  • More experiments are required to validate the proposed method such as applying it on different models such as ResNet with standard datasets.

Answer:

Yes. I think that it is necessary to have more experimental results for the next work. However, as a typical type of the pre-trained models, I think that Although there are many popular pre-trained models such as ResNet, I think that it is suitable to select AlexNet as a representative to validate the effectiveness of the pro-posed method.  The corresponding description have been inserted into the paper. (on pare 6 lines 265-267)

3- comparison with previous methods

Answer:

Yes. The comparison of the proposed method with the previous method of F-measure-based RSRL is described on page 9 lines 340 -341.

  • elaborate more on the novelty of the proposed method

Answer:

Yes. The novelty of this paper is to introduce an internal metric called disentanglement-measure (D-measure), which measuring the degree of disentanglement of final FC layer nodes, and validate the effectiveness of the D-measure in exploring the optimal model for image aesthetics assessment. These description about the novelty can be found on page 3 line 105-107, page 3 lines 111-122, page 15 lines 509-513.

  • elaborate more on the benefit of the proposed method such as in terms of performance

Answer:

Yes. The more benefit of the proposed method is described on page 15 lines 519-520.

  • code , demo are important to be public

Yes. It is possible to open the code and demo, if necessary.

7- More recent  references are required,  I would suggest the following,  update other references

" Review of deep learning: concepts, CNN architecture, challenges, applications, future directions

Answer:

Yes. The above paper is cited on page 7 line 267.

Reviewer 2 Report

This work proposes a new model for image aesthetic assessment and analysis. For this, the author proposes a method based on the recurrent use of CNN networks that allow the extraction of visual characteristics in different regions of an image. The article presents a limited analysis of the state of the art. Additionally, section 2 should be improved. The proposed methodology could have been presented as an algorithm.

There is no clear pattern in the behavior of the RSRL algorithm, which is shown in Fig.4. It is not explained why 30 iterations have been chosen for this model.

The analysis in Fig.7 is very confusing. Must be rewritten (Pg.9.lines 331-338)

The authors mentioned that they analyzed 300 images from the CUHK-PQ dataset. However, there is no performance report on the dataset, except as discussed on page 13 (line 449)

Author Response

Comments and Suggestions for Authors

This work proposes a new model for image aesthetic assessment and analysis. For this, the author proposes a method based on the recurrent use of CNN networks that allow the extraction of visual characteristics in different regions of an image. The article presents a limited analysis of the state of the art. Additionally, section 2 should be improved. The proposed methodology could have been presented as an algorithm.

Answer:

Section 2 defined a new metric called D- measure, and explained the computation produce regarding D-measure. I think that it is difficult to present this produce as an algorithm. I insert a flowchart to explain this produce, which is shown as fig.2 on page 4

There is no clear pattern in the behavior of the RSRL algorithm, which is shown in Fig.4. It is not explained why 30 iterations have been chosen for this model.

Answer:

Yes, there is no clear pattern in the behavior of the RSRL. The optimal model is determined from many models retrained by RSRL based on the expression (8). 30 iterations in Fig.4 is only an example for showing how to select the optimal model from the retrained models. I insert the explanation regarding 30 iterations on page 8 lines 306-307.

The analysis in Fig.7 is very confusing. Must be rewritten (Pg.9.lines 331-338)

Answer:

I have rewritten the part about the analysis in Fig.7, which can be seen on page 10 lines 351-365.

The authors mentioned that they analyzed 300 images from the CUHK-PQ dataset. However, there is no performance report on the dataset, except as discussed on page 13 (line 449)

Answer:

Yes, the performance report has not been done yet. It is necessary to complete for the next work.

Reviewer 3 Report

In this paper, the authors propose a novel deep structure for image aesthetic assessment. Unfortunately, the manuscript is difficult to read and follow. The reviewer's comments are the followings:

1.) The abstract goes into very specific technical details and it is unclear what the main contribution is. 

2.) The structure of the paper is not ideal. The authors omit the literature review or related works section. Mentioning 13 related work is very little in this field. Namely, there are many deep learning based approaches in IAA. First, the authors should mention that the related fields of IAA are dominated by DL models. For example, the authors could cite image quality assessment (Paper: No-reference image quality assessment with convolutional neural networks and decision fusion, 2022). Next, some non deep learning based IAA approaches and the related deep learning based ones. There are many in the literature: https://dblp.org/search?q=image+aesthetics .

3.) Line 148-169: A pseudocode would be welcomed.

4.) If I am correct, Figure 2 has also appeared in a conference publication of the authors. It would be correct to highlight in the introduction, that this journal paper provides more insight, improvements, etc. to the previously published conference paper.

5.) Eq. 6-11 are unaesthetic. Please format them carefully in Latex.

6.) Line 252-284: This type of description of CNN architectures are probably not ideal. A figure would be a better solution...

7.) A comparison to other state-of-the-art methods is missing from the manuscript.

8.) More sample images from the applied database would be welcomed.

9.) Some cases where the network fails to predict the aesthetics would be great in the manuscript. Moreover, a short description of typical error cases would be nice in the manuscript.

10.) The authors use many abbreviations. A list of abbreviations at the end of the manuscript would be helpful for the readers.

11.) A data availability statement with links would be also helpful for the readers, since the authors use publicly available databases.

Author Response

Comments and Suggestions for Authors

In this paper, the authors propose a novel deep structure for image aesthetic assessment. Unfortunately, the manuscript is difficult to read and follow. The reviewer's comments are the followings:

  • The abstract goes into very specific technical details and it is unclear what the main contribution is.

Answer:

I think that I present the main contribution of the paper in the abstract, which is as the following.

An internal metric called disentanglement-measure (D-measure), which reflect the disentanglement degree of the final layer FC nodes of convolutional neural network (CNN) is introduced. By combining F-measure with D-measure to obtain a FD measure, an algorithm of determining the optimal model from many photo score prediction models generated by CNN-based repetitively self-revised learning (RSRL) is proposed. The experimental results show that the proposed method is helpful in improving the efficiency of determining the optimal model.

  • The structure of the paper is not ideal. The authors omit the literature review or related works section. Mentioning 13 related work is very little in this field. Namely, there are many deep learning based approaches in IAA. First, the authors should mention that the related fields of IAA are dominated by DL models. For example, the authors could cite image quality assessment (Paper: No-reference image quality assessment with convolutional neural networks and decision fusion, 2022). Next, some non deep learning based IAA approaches and the related deep learning based ones. There are many in the literature: https://dblp.org/search?q=image+aesthetics .

Answer:

I cited the mentioned paper and other papers in the introduction (page 2 lines 81-89)

3.) Line 148-169: A pseudocode would be welcomed.

Answer:

I think that it is difficult to present this produce using pseudocode. I insert a flowchart to explain this produce, which is shown as Fig.2 on page 4.

4.) If I am correct, Figure 2 has also appeared in a conference publication of the authors. It would be correct to highlight in the introduction, that this journal paper provides more insight, improvements, etc. to the previously published conference paper.

Answer:

This figure is an improved version of the previous conference publication. Please confirm. On the other hand, the related work about the author had been described in the introduction (page 2 lines 95-107)

5.) Eq. 6-11 are unaesthetic. Please format them carefully in Latex.

Answer:

The word format is used by the paper. Accordingly, it is difficult to format Eq. 6-11 in Latex. I am sorry.

6.) Line 252-284: This type of description of CNN architectures are probably not ideal. A figure would be a better solution...

Answer:

Yes, I think so. However, it is difficult for the figure to represent the details of designed CNN architectures. So, I chose this type for description.

7.) A comparison to other state-of-the-art methods is missing from the manuscript.

Yes, I think so. I will do this work for the next step.

8.) More sample images from the applied database would be welcomed.

Yes. I will present more comprehensive results and analysis in the next paper.

9.) Some cases where the network fails to predict the aesthetics would be great in the manuscript. Moreover, a short description of typical error cases would be nice in the manuscript.

Answer:

Yes. I add a example of incorrect prediction, and take a brief analysis (page 14)

10.) The authors use many abbreviations. A list of abbreviations at the end of the manuscript would be helpful for the readers.

Answer:

Yes, I add a list of abbreviations at the end of the manuscript.

11.) A data availability statement with links would be also helpful for the readers, since the authors use publicly available databases.

Answer:

I will try to open the data source used in the paper.

Round 2

Reviewer 1 Report

The authors have addressed the comments but more experiments are required to add in future work.

Author Response

Comments and Suggestions for Authors

The authors have addressed the comments but more experiments are required to add in future work.

Answer:

Thank you for your constructive comments. I add the description about the more experiments in the future work. (page 15 lines 522-525, lines 542-543)

Reviewer 2 Report

All comments have been resolved in the review version

Author Response

Comments and Suggestions for Authors

All comments have been resolved in the review version

Answer:

Thank you for your constructive comments.

Reviewer 3 Report

Eq. 6-11 are still unaesthetic. It is peculiar to me to omit completely the related work section or subsection in such a hot research field... The authors could structure the manuscript more logically. Otherwise, the authors present enough experimental results to support the conclusion. 

Author Response

Comments and Suggestions for Authors

Eq. 6-11 are still unaesthetic. It is peculiar to me to omit completely the related work section or subsection in such a hot research field... The authors could structure the manuscript more logically. Otherwise, the authors present enough experimental results to support the conclusion.

Answer:

Thank you for your constructive comments. I adjusted the layout of these equations. I hope that they can look a little aesthetic. On the other hand, I add the description about the more experiments in the future work. (page 15 lines 522-525, lines 542-543)

Round 3

Reviewer 3 Report

I think the manuscript is an good shape now and suitable for publication.